

# Insights into the morphology of symbiotic shrimp eyes (Crustacea, Decapoda, Palaemonidae); the effects of habitat demands

Nicola C. Dobson[1], Magnus L. Johnson[1] and Sammy De Grave[2]

[1] Centre for Environmental and Marine Sciences, University of Hull, Scarborough, United Kingdom
[2] Oxford University Museum of Natural History, Oxford University, Oxford, United Kingdom

## ABSTRACT

Morphometric differences in the optical morphology of symbiotic palaemonid shrimps can be observed among species symbiotic with different host organisms. Discriminant functional analysis revealed three distinct groups within the species examined. Of these, bivalve symbionts appear to have an eye design that is solely unique to this host-symbiont grouping, a design that spans across multiple genera of phylogenetically unrelated animals. Although some taxonomic effects may be evident, this does not explain the difference and similarities in eye morphology that are seen within these shrimps. Therefore evolutionary pressures from their host environments are having an impact on the optical morphology of their eyes however, as indicated by host-hopping events there ecological adaptations occur post host invasion.

# INTRODUCTION

Symbiotic palaemonid shrimps are widespread and abundant in Indo-West Pacific reefal habitats, characterised by their affinity to form associations with a wide range of taxa. Until recently these shrimps were in the subfamily Pontoniinae. However in a recent phylogenetic study by *De Grave, Fransen & Page (2015)* this subfamily was synonymised with the family Palaemonidae, as were the related families Gnathophyllidae and Hymenoceridae. For the purposes of this investigation and throughout the remainder of this paper, we will refer to this group of shrimps as 'pontoniine shrimps' to avoid any systematic ambiguity. Members of the previously separate families Gnathophyllidae and Hymenoceridae were not included in the present analyses. Within the pontoniine shrimps, an estimated 60–70% (*De Grave, 2001*) are known to form associations with corals, sponges, ascidians, gorgonians, and so on. However this is likely to be an underestimate as the host association remains unknown for several species, but is inferred to be symbiotic due to their morphological similarity to other species. Pontoniine shrimps occur in a wider variety of tropical and subtropical habitats, and are known from deeper water, down to about 2,000 m (*Bruce, 2011*). However, their highest species richness is on tropical coral reefs, down to about 100 m. The most

Corresponding author
Nicola C. Dobson,
n.dobson@biosci.hull.ac.uk

recent catalogue (*De Grave & Fransen, 2011*) lists 602 species, but numerous species have been described since then.

The traditional view of these shrimps as symbionts has recently been challenged for a number of species dwelling in sponges, where diet studies revealed them to be parasites as their stomachs only contained host tissue and spicules (*Ďuriš et al., 2011*). At present it is not known how widespread parasitism is in the group, and we thus refer to them as associates, inferring no trophic interaction with the host.

Morphological adaptation to an associated mode of life has been extensively noted in the taxonomic literature for pontoniine shrimps. Such adaptations include modified pereiopods (*Bruce, 1977*; *Patton, 1994*) in addition to extensive modifications in general body plan and mouthparts (*Bruce, 1966*; *Ďuriš et al., 2011*). Additionally, a range of ecologies are recognised, ranging from internally dwelling in small sized hosts like ascidians (e.g., species of the genus *Periclimenaeus*) to fish cleaning species, dwelling on anemones (e.g., *Ancylomenes* spp.). Despite this wealth of morphological and ecological disparity, few studies have been done linking morphological disparity with ecological constraints. A recent exception to this is the study by *Dobson, De Grave & Johnson (2014)* which examined gross eye morphology across four, broad, lifestyle categories: ectosymbionts, bivalve endosymbionts, non-bivalve endosymbionts, and free-living. Their results clearly demonstrated considerable differences in superficial optical parameters across various lifestyles. In many decapods, vision is thought to be an important feature of their morphology with variations in morphology and structure reflecting ecological habitat demands (*Johnson, Shelton & Gaten, 2000*). Differences in eye size, facet size and interommatidial angle have been observed in many marine species occupying different depths (*Gaten, Shelton & Herring, 1992*; *Johnson et al., 2000*). Eye parameter (EP) has been used by a number of researchers as a measure of determining the equipoise between sensitivity and resolution of different organisms (*Snyder, 1979*; *Stavenga & Hardie, 1989*; *Kawada et al., 2006*). For organisms occupying well-lit habitats EPs of between 0.45 and 1 rad-$\mu$m have been recorded, 1–2 for crepuscular and 2–3 for nocturnal species (*Kawada et al., 2006*), however these values many vary in aquatic organisms due to the different refraction index of water. Pontoniine shrimps are ideal study organisms for the relationship between eye morphology, vision and habitat demands, given their predilection for forming associations with a wide range of taxa.

The current study builds upon this previous work by focussing on and contrasting across actual host identities using a multivariate analytical framework and thus aims to further unravel potential differences in gross optical morphology of pontoniine shrimps.

## METHODS

Optical characteristics of 96 species from 40 genera were examined from collections at the Oxford University Museum of Natural History. A copy of the dataset used in this paper can be accessed in Supplemental Information 1. The work described in this paper was reviewed and approved by the Department of Biological Sciences, Faculty of Sciences ethics committee approval number U053. To understand differences in eye morphology

between host categories, each species was classed into host-symbiont predefined groupings based on their most common host associations (*Bruce, 1994*); i.e., Actiniaria, Ascidiacea, Asteroidea, Bivalvia, Crinoidea, Echinoidea, Gorgonacea, Hydrozoa, Ophiuroidea, Porifera and Scleractinia or considered to be free-living. For all species, eye span (ES), diameter at the base of the eyestalk (DBES), facet diameter (FD) and eye diameter (ED) were measured using a dissecting microscope fitted with an ocular micrometer. To reduce scaling effects ES, DBES and ED were standardised by post orbital carapace length, whilst FD was standardised by eye diameter. A composite variable, ES-DBES (eye span minus diameter at base of eyestalk), was also formulated to provide an indication of eye mobility, the greater mobility of the eyes the larger the value. In addition to the variables measured, eye parameter (EP) was calculated as an outcome of facet diameter ($\mu$m) (FD) and interommatidial angle ($\Delta\varphi$ in radians) using *Snyder*'s (*1979*) equation (Eq. (1)).

$$\text{EP} = \text{FD}\Delta\varphi. \tag{1}$$

Interommatidial angle in radians, used in the calculation of EP, was estimated using an adaptation of *Stavenga*'s (*2003*) formula (Eq. (2)).

$$\Delta\varphi = 2\left(\frac{\text{FD}}{\text{ED}}\right). \tag{2}$$

The presence or absence of the nebenauge (see *Dobson, De Grave & Johnson, 2014*) was also noted and when present the relative size was expressed after standardisation by eye diameter (ED). Our terminology follows *Johnson, Dobson & De Grave (2015)*, who utilised 'nebenauge' for the structure previously referred to under several names.

Eye Parameter (EP) and standardised nebenauge size was compared between hosts using a Kruskal Wallis test in the Statistical Software Package R 3.0.2 as this allowed for *Post Hoc* comparisons (*R Core Team, 2013*), whilst Eye Diameter (ED) was analysed by the means of an ANOVA.

Subsequently, the dataset was analysed with Discriminant Function Analysis (DFA), also known as Multiple Discriminant Analysis (MDA) or Canonical Variate Analysis (CVA). DFA extracts linear combinations of variables (known as roots) which maximise differences amongst a priori defined groups, in this case host categories, with the percentage correctly classified providing a goodness of fit measure, akin to more traditional *P* values.

As DFA requires the number of predictor variables to be fewer than the sample size of the smallest group, a number of host-categories could not be included in the analysis, namely Echinoidea, Hydrozoa, Ophiuroidea and Asteroidea, all of which are relatively infrequently inhabited by pontoniine shrimp. Outliers were identified using within host category linear least-squares regression analysis, using post-orbital carapace length as the independent variable. Individual outliers were corrected by re-measurement (where possible), and only excluded from the final dataset if their values still exceeded 3 standard deviation in residual plots. The final dataset analysed with DFA thus comprised of 83 species, across 7 host categories, as well as free-living taxa. Host categories herein analysed, comprise of Actiniaria (9 shrimp species), Ascidiacea (7), Bivalvia (12), Crinoidea (8), Gorgonacea (7),
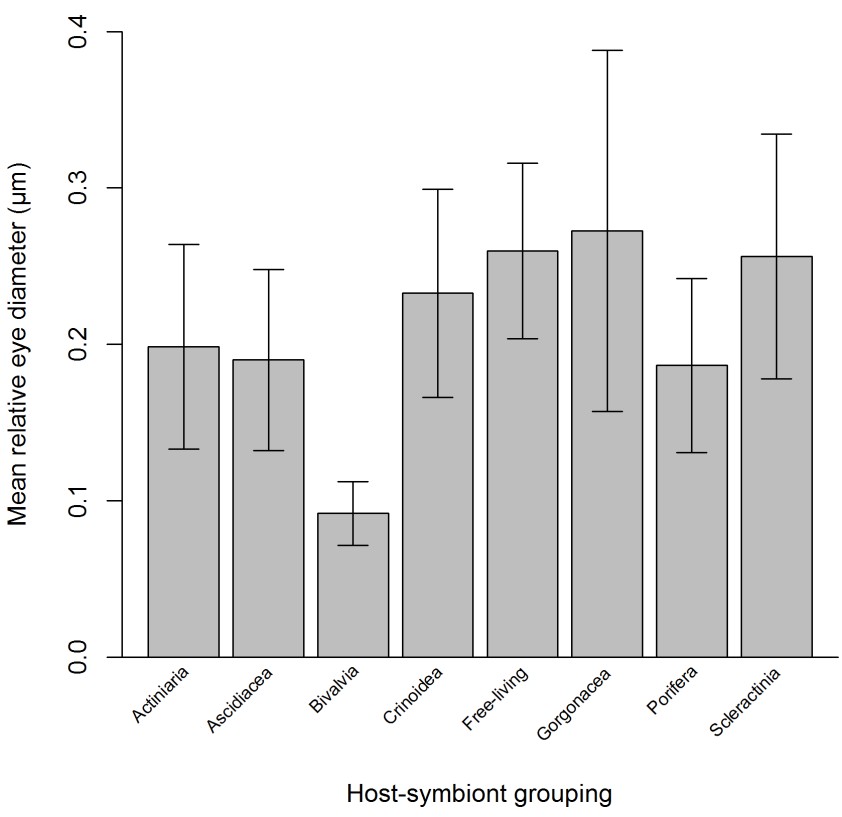

**Figure 1** Mean relative eye diameter (standardised by post-orbital carapace length) for 83 species of Pontoniinae associated 8 host-symbiont groupings.

Porifera (14) and Scleractinia (13). Thirteen micro-predatory species, which are currently considered not to be host associated, i.e., free-living were also included in the analysis, a combination of species living on coral reefs and in seagrass beds. The full species names of the 83 species examined in the DFA analysis are not included on any of the DFA plots however these can be found Supplemental Information 1.

For consistency, statistical analysis of eye size, Eye Parameter and nebenauge was carried out on the reduced dataset.

Prior to DFA, proportions were arcsine-transformed to meet the assumptions for statistical analysis of normality and homogeneity (*Zuur, Ieno & Elphick, 2010*). All DFA analysis was performed in SPSS 18. In all DFA analysis, all variables were entered simultaneously, with the contribution of each variable assessed on the basis of discriminant loadings (structure correlations, rather than discriminant coefficients, as those are considered more valid when interpreting the relative contributions of each variable).

## RESULTS

### Eye size, eye parameter and nebenauge presence

Across all species examined, mean relative ED (Fig. 1) ranged from 0.09 to 0.27, with significantly smaller eyes occurring in bivalve associated species (ANOVA, $F_{7,75} = 9.26$,
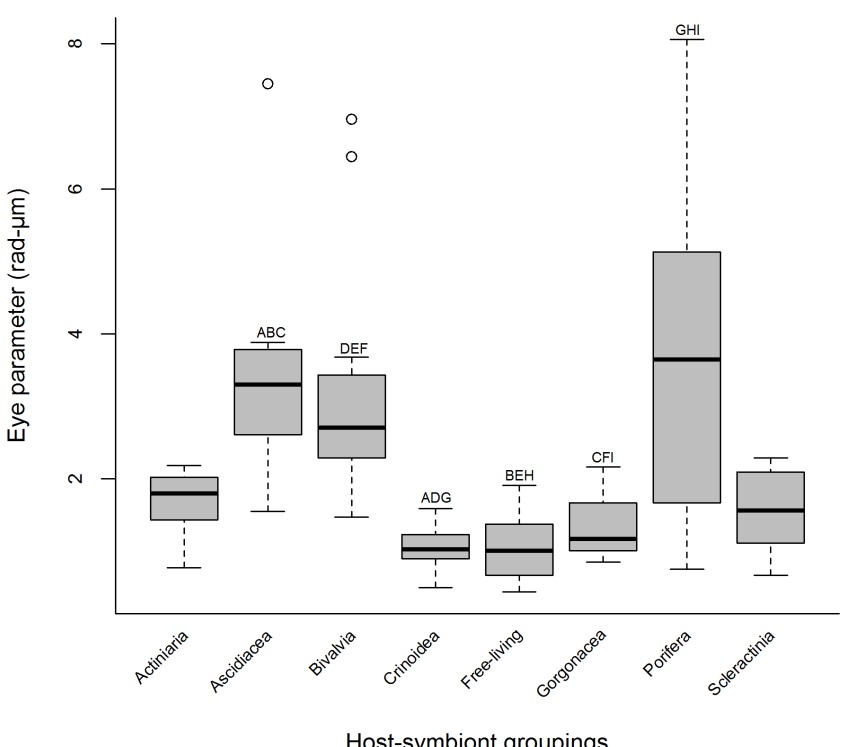

**Figure 2** **Median eye parameter for 83 species of Pontoniinae from 8 host-symbiont groupings.** Significant differences are represented by hosts possessing the same letter A–I (Tukey HSD $P < 0.05$).

$P < 0.001$, Tukey $P = 0.05$). Although the analysis deemed none of the remaining differences to be statistically significant, ascidian ($\overline{x} = 0.19, SD \pm 0.06$) and sponge symbionts ($\overline{x} = 0.19, SD \pm 0.06$) were also found to possess some of the smallest relative EDs whilst gorgonian symbionts ($\overline{x} = 0.28, SD \pm 0.11$) and free-living shrimps ($\overline{x} = 0.26, SD \pm 0.06$) had the largest relative EDs.

Eye parameter (EP) (Fig. 2) ranged from 0.44 to 8.06 rad-$\mu$m, with a significantly larger EP found in ascidian, bivalve and sponge associates (Kruskal Wallis, $H$ (adjusted for ties) = 43.62, $df = 7$, $P < 0.001$, *Post hoc* pairwise comparisons $P = 0.05$). The smallest EP values were found in associates of crinoid, gorgonians and in free-living shrimps. Associates of sea anemones and corals were not significantly different to any other host category in terms of EP (Fig. 2), whilst the widest range of values is present in sponge associates. Although not statistically considered as outliers in within-host category regression analysis, three species exhibited an aberrant EP, all of the genus *Pontonia*. *Pontonia panamica* an ascidian commensal has the largest EP in the dataset (EP = 7.45), whilst *P. mexicana* and *P. pinnophylax* exhibited considerable larger values than other species associated with bivalves.

A significant association was found between the presence/absence of the nebenauge and host category (Chi-squared test, $\chi^2 = 24.777$, $df = 7$, $P < 0.001$). High absence rates of the nebenauge were observed among ascidian, bivalve and poriferan symbionts (Fig. 3), whilst it is prevalent in sea anemone associates and free-living shrimps. However, the relative size

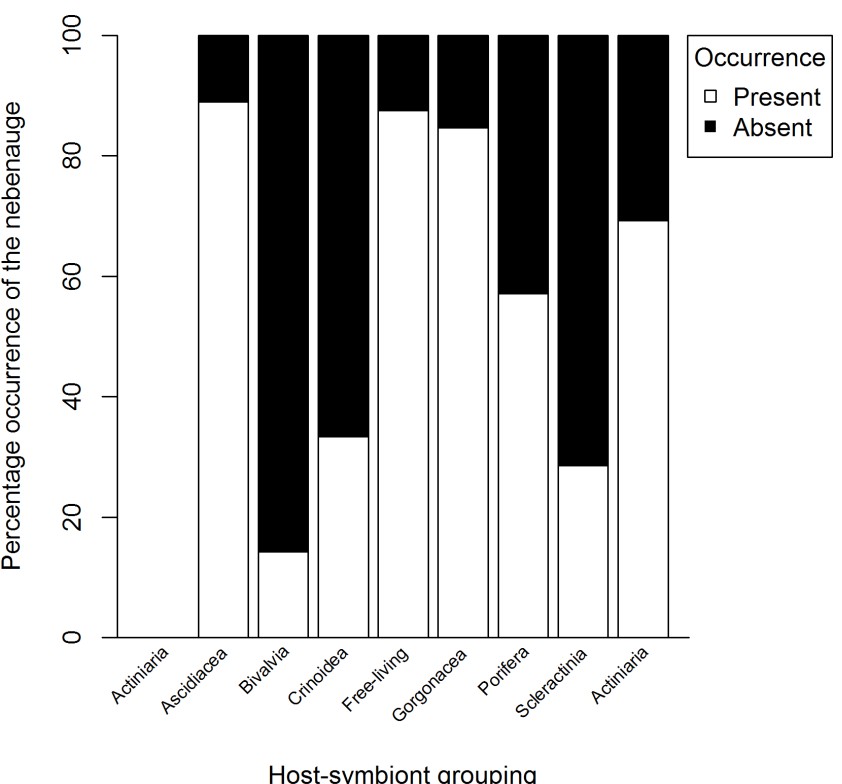

**Figure 3** Percentage occurrence of the nebenauge for 83 species of Pontoniinae from 8 host-symbiont groupings.

**Table 1** Summary statistics for DFA analysis.

|  | Eigenvalue | % of variance | Cumulative % | Canonical correlation | Wilks's λ | P value |
|---|---|---|---|---|---|---|
| Root 1 | 1.436 | 59.1 | 59.1 | 0.768 | 0.194 | <0.005 |
| Root 2 | 0.864 | 35.5 | 94.6 | 0.681 | 0.473 | <0.005 |
| Root 3 | 0.090 | 3.7 | 98.4 | 0.288 | 0.882 | 0.482 |
| Root 4 | 0.040 | 1.6 | 100 | 0.196 | 0.962 | 0.561 |

is not different across host categories (Kruskal Wallis test, $H = 8.93$, $df = 6$, $P = 0.178$), with ascidians excluded as only one species, *Periclimenaeus hecate*, had a nebenaugen.

## Multivariate analysis

Discriminant function analysis revealed only two significant roots (Table 1), which cumulatively explain 94.6% of total variance. Examination of the structure matrix (Table 2) revealed that three variables were highly loaded on to the first root (EP, FD, ED), whilst a fourth variable (ES-DBES) displayed greatest loading on the second function.

A classification matrix indicates that overall 50.6% of shrimp species were correctly classified in respect to their priori defined groups (host classification) (Table 3), but with

**Table 2  Structure matrix of discriminant loadings, with the largest absolute correlation between each variable and any discriminant function indicated by \*.** All variables were entered simultaneously.

|  | Function 1 | Function 2 |
|---|---|---|
| ArcsinFD | 0.808* | 0.482 |
| EP | 0.718* | 0.166 |
| ArcsinED | −0.657* | 0.481 |
| ES-DBES | −0.158 | −0.695* |

**Table 3  DFA Classification matrix, showing number of species correctly and incorrectly classified into a priori defined groups, expressed as a percentage of within group species numbers.**

|  |  | DFA classification | | | | | | | |
|---|---|---|---|---|---|---|---|---|---|
|  |  | Actiniaria | Ascidiacea | Bivalvia | Crinoidea | Non-commensal | Gorgonacea | Porifera | Scleractinia |
| A priori groups | Actiniaria | 22.2 | – | 11.1 | 11.1 | 22.2 | – | – | 33.3 |
|  | Ascidiacea | 14.3 | – | 14.3 | – | – | – | 71.4 | – |
|  | Bivalvia | – | – | 100.0 | – | – | – | – | – |
|  | Crinoidea | 25.0 | – | – | 12.5 | 12.5 | 37.5 | – | 12.5 |
|  | Non-commensal | 7.7 | – | – | 15.4 | 61.5 | – | – | 15.4 |
|  | Gorgonacea | – | – | – | 28.6 | 14.3 | 42.9 | – | 14.3 |
|  | Porifera | 7.1 | – | – | 14.3 | – | – | 78.6 | – |
|  | Scleractinia | 15.4 | – | – | 7.7 | 15.4 | – | 23.1 | 38.5 |

significant variation as to within-group classification. Bivalve associates were 100.0% correctly classified, with a high number also correctly classified for sponge associates (78.6%). Over half of the free-living species (61.5%) were correctly classified to their priori group, with other species classified as sea anemone, crinoid and coral associates. Gorgonian associates correctly classified in 42.9% of cases, with misclassified taxa allied to free-living, coral and crinoid associates. Coral associates correctly classified in 38.5% of cases with species misclassifying as associates of sponges, sea anemones, crinoids and free-living species. Sea anemone and crinoid associates were only 22.2 and 25.0% correctly classified. All ascidian symbionts were found to misclassify, with 71.4% of them misclassified as sponge associates.

When comparing the relative position of the centroids for each host category (Fig. 4) it is obvious, that the eyes of ascidian and sponge associated species are very similar to each other, as are the eyes of crinoid and coral associates, both of which also group with the free-living species. Although broadly similar to the latter grouping, the eyes of gorgonian and sea anemone associates are somewhat divergent as well as divergent to each other, as evidenced by the position of their centroids. Bivalve associates clearly occupy an isolated position, relative to the other host categories.

When plotting only the ascidian associates in the DFA analysis (Fig. 5), a divergent position of *P. panamica* is evident, whilst the other taxa form a loose grouping. The positions of sponge associates (Fig. 6) reveal two distinct, but loose groupings, as well as a

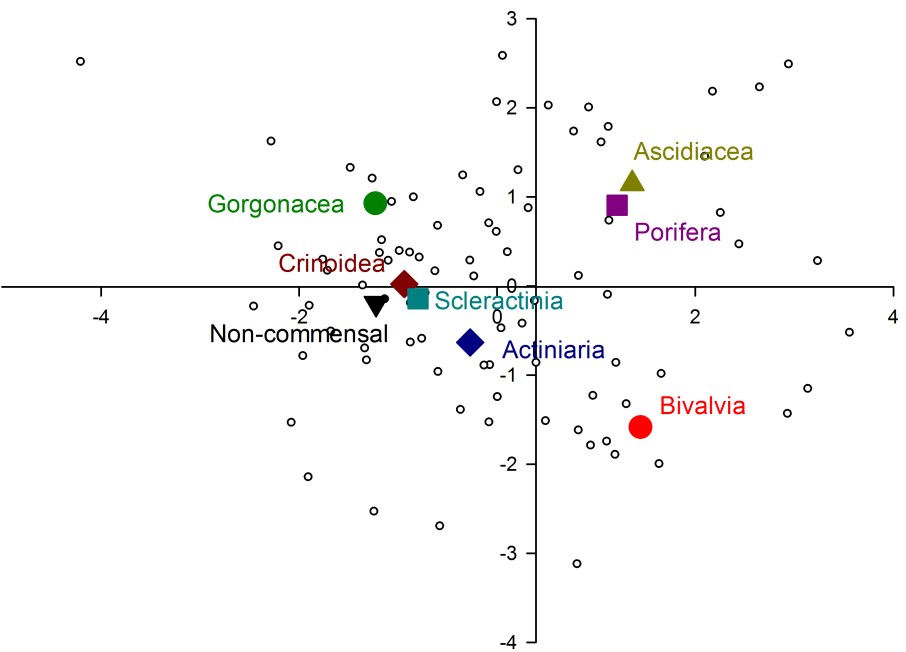

**Figure 4** Morphological variation demonstrated by the DFA scores (first and second root only) of all 83 species of pontoniine shrimps (grey circles) displaying the positioning of the centroids for each of the 8 hosts-symbionts groups.

divergent species, *Thaumastocaris streptopus*. Membership in either of the two groups does not appear to be influenced by phylogeny, as either group contains species belonging to the genera *Typton* and *Periclimenaeus*. The positions of the individual bivalve associates (Fig. 7) reveals a relatively tight grouping, but with an isolated position occupied by *Conchodytes nipponensis*. The positions of individual crinoid associates (Fig. 8) are rather scattered, but with a very isolated position for *Laomenes nudirostris*. A similar scattered pattern is observed for the coral associates (Fig. 9) and the free-living species (Fig. 10). Gorgonian associates also demonstrate this pattern (Fig. 11), but with a significant, isolated position for *Pontonides loloata*. A similar pattern is observed for sea anemone associates (Fig. 12), with an isolated position for *Periclimenes scriptus*.

## DISCUSSION

Multivariate analysis clearly reveals that three distinct eye types are present in pontoniine shrimps, with bivalve associates comprising a type on their own. Sponge and ascidian associates have remarkably similar eyes, to the point that the majority of ascidian associates were misclassified as sponge associates in the analysis. A third eye type is present in a range of ectosymbiotic taxa, associated with sea anemones, gorgonians, corals, crinoids, as well as free-living species.

An examination of the structure loadings reveals that along the first root, both facet diameter (FD) and Eye Parameter (EP) increases, but with a concomitant decrease in eye diameter (ED), whilst along the second root eye mobility (as measured by ES-DBES) decreases. Broadly speaking, the ectosymbiotic and free-living taxa thus have smaller facet

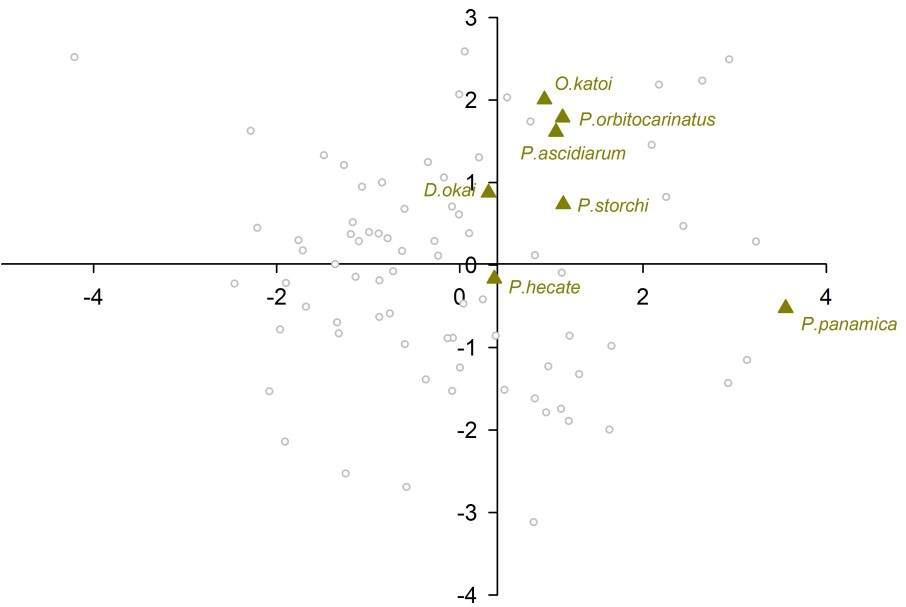

**Figure 5 Morphological variation demonstrated by the DFA scores (first and second root only) of Ascidiacea associates.**

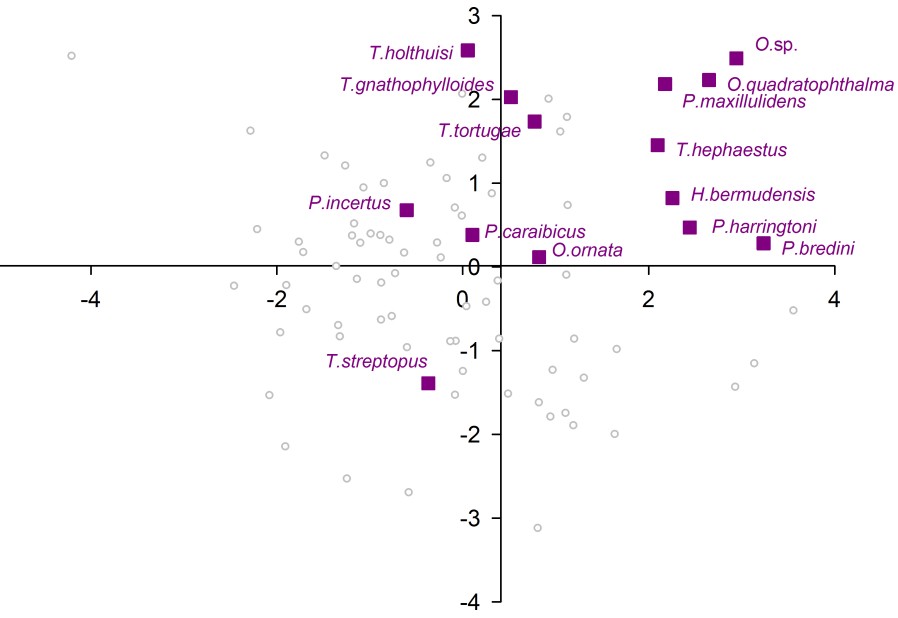

**Figure 6 Morphological variation demonstrated by the DFA scores (first and second root only) of Porifera associates.**

diameters, a lower EP and bigger eyes, than their endosymbiotic counterparts in bivalves, sponges and ascidians. Equally, bivalve associates display more mobile eyes than ascidian and sponge associates, but with roughly similar facet diameter and EP. It should be noted that the relative eye size of bivalve associates is significantly smaller than all other host groupings, this may be as a result of their comparably larger body sizes (e.g., mean average

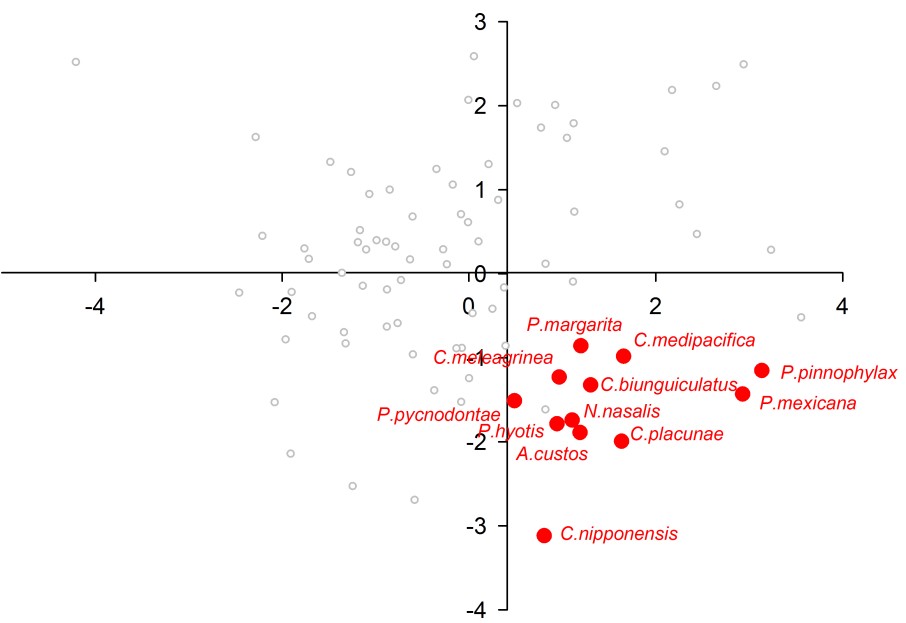

**Figure 7** Morphological variation demonstrated by the DFA scores (first and second root only) of Bivalvia associates.

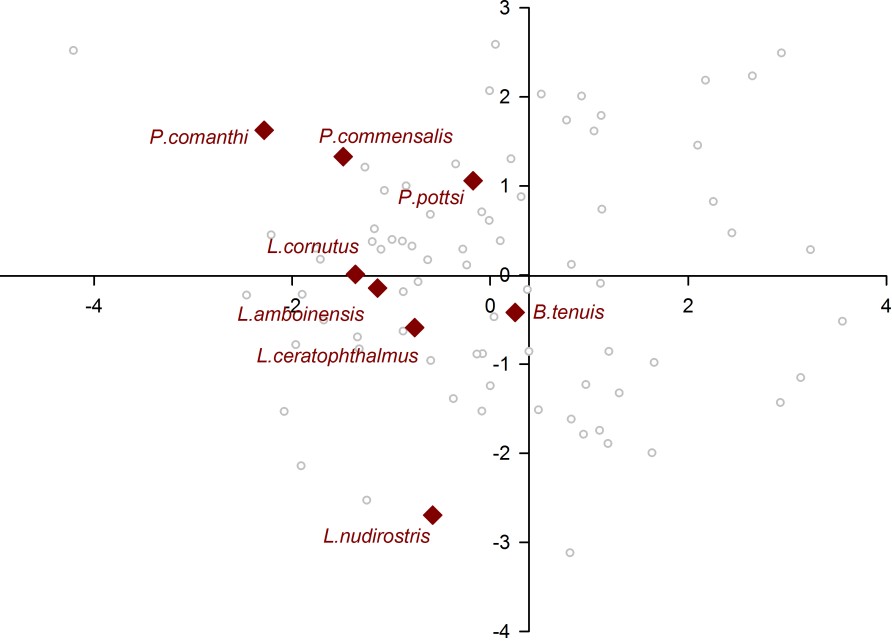

**Figure 8** Morphological variation demonstrated by the DFA scores (first and second root only) of Crinoidea associates.

6.9 mm CL versus 3.0 mm CL for Actiniaria, 2.5 mm CL for Porifera and 1.34 mm CL for Gorgonacea symbionts).

Within deep sea caridean species the nebenauge has been suggested to have an important role in diurnal migrations (*Johnson, Dobson & De Grave, 2015*). The concept that orientation to light is aided by the presence of the nebenauge is further supported by these

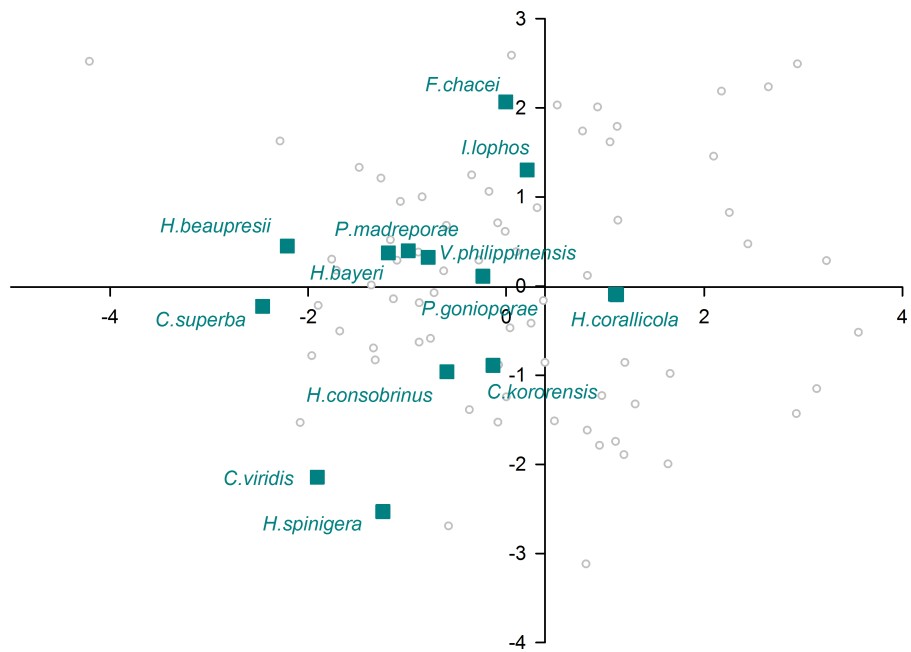

**Figure 9** Morphological variation demonstrated by the DFA scores (first and second root only) of Scleractinia associates.

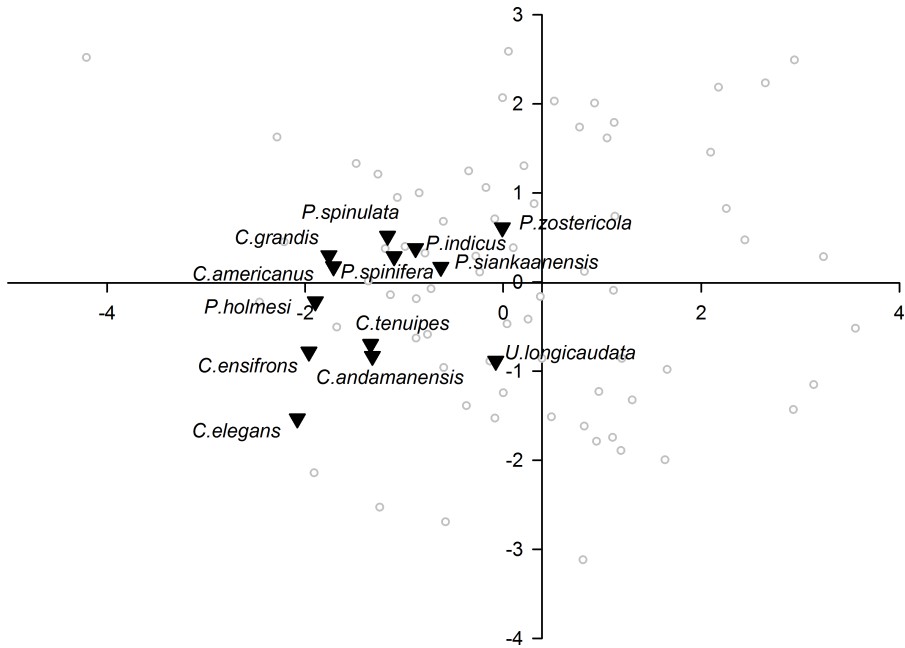

**Figure 10** Morphological variation demonstrated by the DFA scores (first and second root only) of non-commensal species.

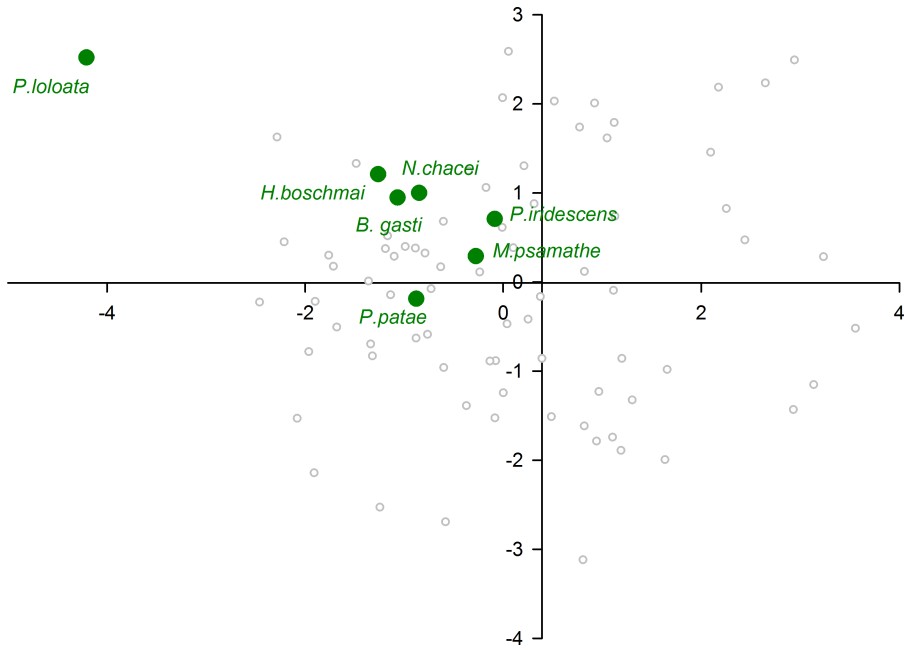

**Figure 11    Morphological variation demonstrated by the DFA scores (first and second root only) of Gorgonacea associates.**

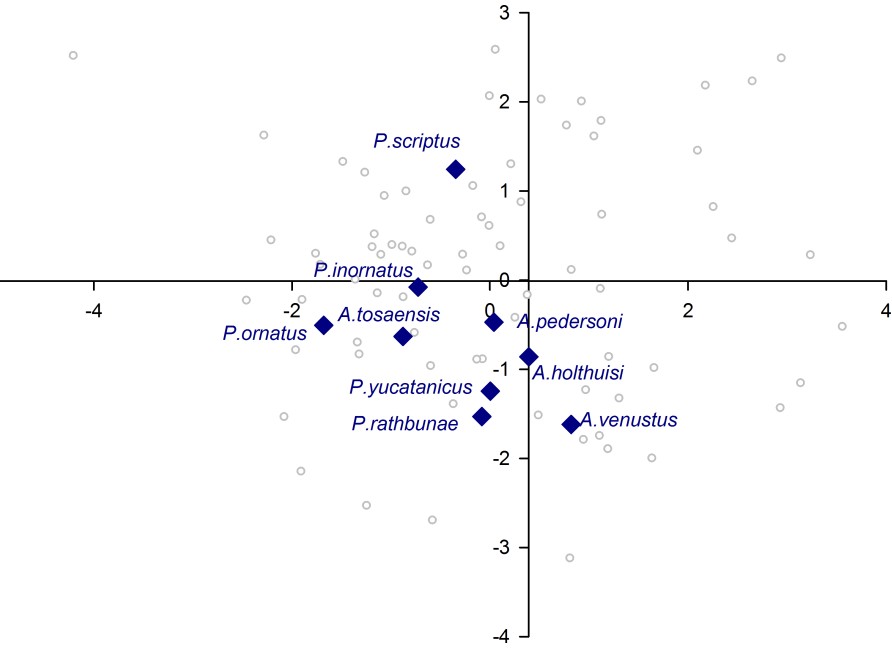

**Figure 12    Morphological variation demonstrated by the DFA scores (first and second root only) of Actiniaria associates.**

results with it being highly abundant within sea anemone, crinoid, free-living and coral associates. However for bivalve, ascidian and sponge associates both diurnal migrations and orientation to light would be of little significance for species with an endosymbiotic mode of life.

This result is not surprising, given the clear relationship between gross eye morphology of pontoniine shrimps and life style already demonstrated in *Dobson, De Grave & Johnson (2014)*. Therein, based on a range of optical parameters, the eyes of free-living and ectosymbiotic species were found to be very similar, and clearly different from both types of endosymbiotic species considered, bivalves and non-bivalve associates. Further, bivalve endosymbionts exhibited an intermediary group between free-living/ectosymbionts and non-bivalve endosymbionts, potentially linked to their presumed more active lifestyle, with bivalve associated documented to move hosts in search of a mate (*Baeza et al., 2011*).

Whilst the relationships between optical parameters and lifestyle in *Dobson, De Grave & Johnson (2014)* appears clear-cut and supported by the present analysis, by including actual host identity, rather than lifestyle in the current analysis, a number of surprising findings emerge.

The eyes of ascidian associated species emerges as being remarkable similar to the eyes of sponge associated species, to the point that the majority of a priori classified species in this group were misclassified as sponge eyes by the multivariate analysis. This is herein interpreted being likely a significant signal of phylogenetic constraint, as four out of the seven species in this host category belong to a primarily sponge dwelling genus, *Periclimenaeus* (see below) with generally conservative eye morphology, potentially indicative of recent host switching event(s). Two further species in this host category, phylogenetically unrelated to *Periclimenaeus*, *Dactylonia okai* and *Odontonia katoi* are thought to be closely related species (*Fransen, 2002*), but with significantly different gross eye morphology. *Dactylonia okai* possesses stout triangular shaped eyes, whereas the eyes of *Odontonia* species are small and hemispherical (*Fransen, 2002*). Whilst *D. okai* and *O. katoi* are found living within large solitary ascidians, species of *Periclimenaeus* are found living within both ascidians and sponges. Species such as *Periclimenaeus orbitocarinatus* and *Periclimenaeus ascidiarum* live in association with compound ascidians that are structurally similar in morphology to the canals of sponges occupied by, for example, *Periclimenaeus maxillulidens*. The structural similarity in hosts between the symbionts of compound ascidians and sponges could be a plausible explanation for the high misclassification of ascidian symbionts to sponges. Two species were misclassified as either a sea anemone or bivalve associate. Although DFA does not provide information on individual classified species, it is evident from Fig. 5 that *P. panamica* is the species misclassified as a bivalve associate. The genus *Pontonia* comprises of 11 species (*De Grave & Fransen, 2011*) and is morphologically very conservative. Although the host for one species, *P. longispina*, is not known, the majority of species associate with bivalves in the families Pinnidae and Pteriidae, whilst one poorly known species *P. chimaera*, is thought to be an associate of large gastropods of the genus *Strombus*. *Pontonia panamica* is the only species to associate with ascidians, the solitary species *Ascidia interrupta* in the eastern Pacific. Although *Marin & Anker (2008)* speculate that a host switch to ascidians occurred early on in the

evolutionary history of this genus, the retention of essentially a "bivalve" eye is perhaps indicative of a more recent host switching event. However, on balance the differences in eye morphology between the phylogenetically not related genera herein analysed as ascidian associates suggests that despite occurring in a similar host environment, their enclosure inside ascidians has not provided pressure on their eyes to become optically similar. As to whether this lack of overall evolutionary pressure is imparted by distinctive host morphologies (compound, solitary) or habitats (intertidal, subtidal) or indeed is determined by differential behavioural attributes (social biology) of the associates themselves remains unclear.

Notwithstanding their close similarity to ascidian associate eyes, the eyes of sponge associated species appear to be quite uniform, with the majority being correctly classified in their a priori defined host group, but seemingly forming two distinct subgroups in the analysis, in addition to the outlying *T. streptopus*. We infer here that the classification into two subgroups is putatively related to host morphologies, as sponge species exhibit a discrete and distinct range of canal sizes. Space partitioning, as well as individual host selection is indeed known to play a significant role in the sponge-dwelling gambarelloides group of *Synalpheus* (*Duffy, 1992*; *Hultgren & Duffy, 2010*; *Hultgren & Duffy, 2012*). The speculation that canal sizes of the host may play a significant role in optical acuity of pontoniine species, can however not be substantiated, as the host range of most species remains unknown, with even the identity of many hosts simply not being known. For instance, for many species of *Periclimenaeus*, a primarily sponge associated genus, the hosts are not known (*Bruce, 2006*). Of particular interest are the three ectosymbiotic species included in this primarily endosymbiotic group in the present analysis, *T. streptopus*, *Periclimenes harringtoni* and *Periclimenes incertus*. *Thaumastocaris streptopus* is an Indo-Pacific species, which dwells in the central atrium of vase-shaped sponges like *Siphonochalina* and *Callyspongia* (see *Bruce, 1994*). Based on the present suite of optic parameters, this species does not cluster with the rest of the sponge associates. Although *Ďuriš et al. (2011)* consider the species to be parasitic, in common with several other sponge associates, the isolated position of the species in the present analysis, combined with their asymmetrical first pereiopods and a segmented carpus (both unique within the family) is indicative perhaps of a different behavioural niche. The Indo-Pacific, *P. incertus* dwells on the outside of a variety of sponges, and clusters reasonably close to the other sponge associates in the present analysis, potentially indicative of similar relationship to the host, if external. The Caribbean *P. harringtoni* dwells in the atria of *Neofibularia nolitangere* and based on the optical parameters studied herein, appears to have an eye structure very similar to that of endosymbiotic species, potentially an example of habitat driven adaptation, despite the significant difference in position on the host.

The sea anemone associates included in the present analysis, fall into four ecological/systematic groups, *Ancylomenes* and three different species groups of *Periclimenes*. *Ancylomenes* species are on the whole considered to be fish cleaners, who only utilise the sea anemone as an advertisement for their services to client fish (*Huebner & Chadwick, 2012*). It should be noted that this is potentially a generalisation, as direct observation of fish cleaning behaviour is not available for all species, with this information lacking
for one species herein included *A. tosaensis*. *Periclimenes yucatanicus* and *Periclimenes rathbunae* are active large bodied species, associated with a variety of sea anemones in the Caribbean. Fish cleaning has not been observed for either species, with *Limbaugh, Pederson & Chace (1961)* considering *P. yucatanicus* a fish-cleaning mimic. *Periclimenes ornatus* and *P. inornatus* belong to the same species complex, and are smaller bodied species which hide in between the tentacles of a variety of Indo-Pacific sea anemones. Finally, *P. scriptus*, a Mediterranean and subtropical Northeast Atlantic species which is not phylogenetically closely related to the other two groups, is an active species, associated with long tentacle sea anemones, with no known fish cleaning behaviour. With the exception of *P. scriptus* (see below) these species exhibit a scattered grouping in the DFA analysis, and as a group have a low percentage correctly classified, at 22%. It thus appears that despite their broad ecological niche similarity as sea anemone associates, insufficient convergent pressure on their optical parameters is noted, indicative of differential usage of their eyes.

In contrast to sea anemone associates, coral associates exhibit a reasonable level of correctly classified in the DFA analysis, at 38.5%, despite the large variety of host morphotypes involved in this association. Several species including *Coralliocaris* spp., *Harpilius* spp. and, *Harpiliopsis* spp. are associated with branching corals of the families Pocilloporidae and Acroporidae. Other species in this group are associated with corals that extend their polyps during the day, depending of the species of coral these polyps can be with short or long in size. Examples of species inhabiting these hosts include *Hamopontonia corallicola* found within the short polyps of *Goniopora* and *Cuapetes kororensis* that dwells within the long polyps of *Heliofungia actiniformis*. Morphologically heavily modified taxa are also present in this group, such as the laterally flattened *Ischnopontonia lophos* which moves between the corallites of *Galaxea*. It thus appears that the habitat and/or behaviour in the case of coral associates is a significant driver in optical parameters, akin to the free-living species, which had an approximately similar level of correctly classified species (53.8%). However, in contrast to free-living taxa, which are considered to be micro-predators, several of the coral associates are potentially parasites (*Stella et al., 2011*). The common functionality of their optic parameters (to a degree) remains unclear, although it is known that several species, e.g., *Coralliocaris* defend their coral host against predators (*Marin, 2009a*; *Stella et al., 2011*), perhaps necessitating the need for similar optical acuity to free-living micro predators.

Bivalve associates exhibited a 100% correct classification in the DFA analysis, although with reasonable scatter in the scatter plot, and a significant outlier (*C. nipponensis*). Yet the group consists of several genera, including *Conchodytes* and *Anchistus*, which are phylogenetically distant (*Kou et al., 2014*). Furthermore these species can be differentiated by general bauplan morphologies, ranging from relatively unspecialized (*Anchistus* and *Paranchistus* for example) to dorso-laterally compressed (e.g., *Conchodytes*) (*Bruce, 1981*; *Fransen & Reijnen, 2012*). Their phylogenetic distance is evidence of multiple host invasions (*Kou et al., 2014*), but the present analysis reveals considerable convergence in optical parameters, indicative of profound habitat induced restraints.

A number of species occupy isolated positions within their respective groups, notably *P. loloata*, *P. scriptus*, *C. nipponensis* and *L. nudirostris*. Although we cannot discount

variation in optical parameters of individual eyes, which may have lowered the percentage correctly classified and induced a higher degree of scatter, two species are worthy of further discussion. The corneal part of the eye of *Laomenes* species is characterised by an apical papilla (see illustrations for several species in *Marin, 2009b*) which contains functional facets, but which are somewhat different in shape to facets elsewhere on the cornea. The relative size as well as the exact position of the papilla has been used as a minor taxonomic character to differentiate between species (*Marin, 2009b*). However, it is known that a large degree of infra-specific variation is present, which unquestionably would influence some of the herein included optical parameters. *Periclimenes scriptus* appeared isolated within the sea anemone grouping however due to the small size of the specimen (CL 1.25 mm) it is possible that this animal was not fully mature as ovigerous females have a reported CL of 5.0 mm (*Ďuriš et al., 2013*).

## CONCLUSIONS

Overall, our analysis demonstrates that there is a significant evolutionary pressure of the host environment on the optic parameters of associate shrimp species, with in many cases congruence being evident between phylogenetically unrelated taxa. This is especially evident in bivalve and sponge associates, and to a lesser extent in other host taxa. This result is in sharp contrast to the disparate morphology of many other body parts of pontoniine shrimps, with significant variation in mouthparts, pereiopods and even general body shape between genera, inhabiting the same host. At the same time, evidence emerges from the optical analysis of recent host switching events in certain lineages, where the optical parameters have not evolved to a communality yet, especially in the genera *Periclimenaeus* and *Pontonia*, where taxa living in different hosts appear to retain a close optical similarity to those living in other taxa.

## ACKNOWLEDGEMENTS

We would like to thank Dr. Sue Hull for her valuable suggestions with regards to the analysis and comments on previous versions of the manuscript. We would also like to thank both reviewers for their valuable comments on the manuscript.

### Funding

Access to the Oxford University Natural History Museum collections was facilitated through a St John's Summer Scholarship to Magnus L. Johnson in 2011. The funders had no role in study design, data collection and analysis, decision to publish, or preparation of the manuscript.

### Grant Disclosures

The following grant information was disclosed by the authors:
St John's Summer Scholarship.

## Competing Interests

Magnus Johnson is an Academic Editor for PeerJ.

## Author Contributions

- Nicola C. Dobson conceived and designed the experiments, performed the experiments, analyzed the data, contributed reagents/materials/analysis tools, wrote the paper, prepared figures and/or tables, reviewed drafts of the paper.
- Magnus L. Johnson analyzed the data, reviewed drafts of the paper.
- Sammy De Grave analyzed the data, contributed reagents/materials/analysis tools, wrote the paper, prepared figures and/or tables, reviewed drafts of the paper.

## Ethics

The following information was supplied relating to ethical approvals (i.e., approving body and any reference numbers):

The work described in this paper was reviewed and approved by the Department of Biological Sciences, Faculty of Sciences ethics committee: approval number: U053.

## Data Availability

A dataset listing the taxa authority for each species can be found at ResearchGate: 10.13140/RG.2.1.3753.3844.

## Supplemental Information

Supplemental information for this article can be found online at http://dx.doi.org/10.7717/peerj.1926#supplemental-information.

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
