# Peer review of "Insights into the morphology of symbiotic shrimp eyes (Crustacea, Decapoda, Palaemonidae); the effects of habitat demands"

_PeerJ, doi:10.7717/peerj.1926_

## Round 0.1 · original submission · Minor Revisions

· Academic Editor

Minor Revisions

Both reviewers were generally positive about this paper, but both also had some comments in common. In particular, there are some not-major taxonomic issues and corrections in places (such as Palaemonidae) that need to be addressed.

Reviewer 1 ·

Basic reporting

Article well meets all PeerJ standards. The only inconsistency I see in not detailed presentation of species in the DFA diagrams (Figs 5-12) where generic names are to be shown nor abbreviated (see below - General comments). That may easily be improved.

Experimental design

No Comments.

Validity of the findings

No Comments.

Additional comments

This is a straight forward analysis of eye structures and biometry of symbiotic shrimps in relation to their host animals’ phyla. The manuscript (MS) is generally well structured, and the diagrams informative. The methodology of analyses, presented results and their interpretation in the Discussion are generally correct, the obtained results innovative. I found most of the results very interesting and useful for their actual, as well as future, interpretations in the light of symbiotic relations and systematic affiliations of analysed shrimps. The provided results show a nice series of convergences in unrelated taxa explained by occupying a similar habitat, together with interesting divergences in eye characteristics within species of the same host type or shrimps’ systematic affiliations. I see possibilities in future expression of the presented results via molecular phylogenies of shrimps and more detailed division of host affiliations.
Together with very good contribution to the knowledge on crustacean morphology and ecology, the MS contains some inconsistencies and errors. Most of my criticisms deals with formal details which might easily be modified during a revision.
Main remarks are presented below, most minor ones are indicated immediately in the attached pdf of the revised MS. The numbers in square brackets refer to sequential line numbering.

Taxonomy
[1, 24, …] Throughout the paper (also in the title), is given that the object of the paper deals with shrimps of the palaemonid subfamily Pontoniinae. This traditional group of symbiotic shrimps was, however, recently confirmed as polyphyletic, and synonymised with the higher taxon, the family Palaemonidae (De Grave, Fransen & Page, 2015: PeerJ, 3: e1167). The ‘Pontoniinae’ thus no more exists as valid taxon. That is the family Palaemonidae. I see no principal problem with analysing the ‘pontoniines’ as a ‘historical group’ of taxa, despite most likely composed of different evolutionary lineages. But on my opinion, authors then would at least formally point on the actual position of the analysed shrimp assemblage in the Introduction, and interpret obtained results in the Discussion respecting that evidence. The term ‘pontoniid shrimps’ (= Pontoniidae) [9] is incorrect as the suffix ‘-dae’ refers to the family level, while only the subfamily (-nae) existed before.

[137] taxa authorisations are given in the text in cases incorrectly when author’s name and year, following the ICZN code, must be written in parenthesis [also – 166, 167, 213, 268, 288, …?].

[38] The terms like ‘Periclimenaeus spp. Borradaile, 1915’ [also 39-40, 283-284, 287, 290] are incorrect as the given authorisations refers only to generic names, not to the species composition (‘spp.’). Correct is, e.g., ‘species of the genus Periclimenaeus Borradaile, 1915’.

Terminology
[68, 102, and throughout MS] term ‘non-commensals’ are defined [102] as not host associated micropredators. Commensalism is a distinct kind of symbiosis. Non-commensal animals might, in particular, also be symbionts, despite not living on/in their symbiotic partner (i.e., fish cleaners). More correct is: non-symbiotic, non-associated, free-living.

[80, and throughout MS] term ‘nebenauge’. Despite referred [80] to Dobson et al. (2014), nor there, nor here it is explained why clearly the German term “Nebenauge” (literally = accessory eye) is applied in English text for the organ of still quite enigmatic organ on crustacean eyes, but which already was adequately called in English literature as (again literally) “accessory eye”, “accessory eye spot”, etc. The usage of the term “nebenauge” does not more specify the organ (this is not the only ‘additional eye’ in arthropods), and does not provide any additional value to it. I see no reason for that and prefer using any of the English terms above, despite (or because) being equally indefinite, provisional.

Methods, Results
[61] ‘96 species from 40 genera were examined’ … The MS is not provided with an directly accessible list of the species analysed (only in Supplemental dataset). That is a serious obstacle for readers as the Figs 5-12 showing the results of the DFA multivariate analysis for each of the 8 ecological assemblages of shrimps are confusing; in all of them (but Fig. 12) the sets of taxa presented contain species of different generic names but with the same abbreviated capital letter... Most analysed species appear only in those diagrams, thus, nowhere with a full name in the main text (except Suppl. Dataset, but not alphabetically ordered there). It is duty of the authors to explain all abbreviated terms. In the diagrams, full, not abbreviated, species names are preferred for reader’s convenience.

[143-144] Define ‘a priori ..[affiliation of shrimps].. in respective host classification’. Why, e.g., ascidian symbionts ‘were found widely misclasified as sponge associates’ [150-151], but not vice versa (i.e., spongobionts as ascidiobionts)?

Supplemental dataset
The main (supplemental) dataset is not properly indicated in the MS. A ‘dataset’ is mentioned in the paper [86, 97, 98, 105], and only the ’dataset (7.45)’ (?) [129] leads to the Supplemental spreadsheet of primary and secondary data. A reference to it is expected in the beginning of the Methods chapter.

The dataset is not alphabetically ordered by any column, thus, difficult to search for individual data. Presented as an Excel table, reader can, of course, arrange them itself, but original listing by species in alphabetic order is preferred. “Typton holthuisi sp. Nov.” [sic.] is surely not valid term here, Periclimenes scriptus (J) ¬– error or not explained “J”. Chernocaris placunae is now Conchodytes placunae. (J), sp., sp. nov., undescribed – those terms are not parts of species names, thus, must not be written in italics (same in the figures – DFA diagrams). Abbreviations in headings are not explained (or not referred to Methods chapter). The presenting of primary data measurements in mm (columns: Cl, ES, DBES, etc.) with more than 2 decimal digits are unsuitable.

References
Works of the following first authors - Bruce; De Grave; Marin – are not presented in alphabetical order.

Reviewer 2 ·

Basic reporting

The submission meets the standards of PeerJ.
In a recent publication (De Grave, Fransen & Page, 2015) this subfamily has been synonymized with Palaemonidae. I would suggest to replace references to this subfamily with 'symbiotic palaemonids' as far as appropriate. This however could be problematic with regards to the free-living species.

Experimental design

No Comments.

Validity of the findings

There is an error with regards to the host group of one of the species in the analyses. Balssia gasti has never been recorded as a symbiont of Scleractina. The dominant host for this species are Gorgonacea. In rare cases it has also been found on sponges, but never on Scleractinia. It is unfortunate, but the analyses have to be repeated with this error corrected.

Additional comments

Some minor additinal corrections have been indicated in the pdf.

With regards to the supplementary file I have the following comments:

- Balssia gasti is indicated as an symbiont of Scleractinia. To my knowledge the dominant hosts for Balssia gasti belong to the Gorgonacea. The species has incidentally been found on sponges, but never on Scleractinia. I am afraid all analyses have to be repeated with this correction.
- 'Harpiliopsis beaupressi' should be 'Harpiliopsis beaupresii'.
- Periclimenes siankaanensis has been transferred to the genus Phycomenes.
- What means the (J) behind Periclimenes scriptus?
- Typton holthuisi has been published by De Grave in 2010. The ‘spec. nov.’ behind this name can be omitted.
- Chernocaris placunae has been transferred to the genus Conchodytes by Fransen & Reijnen (2012).

Annotated reviews are not available for download in order to protect the identity of reviewers who chose to remain anonymous.

---

## Round 0.2 · Minor Revisions

· Academic Editor

Minor Revisions

The manuscript has been revised well, and the authors have answered almost all of the reviewers' comments adequately. I have also been through the submission, and have only noticed a few small corrections (see attached PDF) that need taking care of.

However, there is one reviewer comment that I feel needs to be addressed still:

[61] ‘96 species from 40 genera were examined’ … The MS is not provided with an directly accessible list of the species analysed (only in Supplemental dataset). That is a serious obstacle for readers as the Figs 5-12 showing the results of the DFA multivariate analysis for each of the 8 ecological assemblages of shrimps are confusing; in all of them (but Fig. 12) the sets of taxa presented contain species of different generic names but with the same abbreviated capital letter... Most analysed species appear only in those diagrams, thus, nowhere with a full name in the main text (except Suppl. Dataset, but not alphabetically ordered there). It is duty of the authors to explain all abbreviated terms. In the diagrams, full, not abbreviated, species names are preferred for reader’s convenience.

Author response: The DFA analysis was repeated to correct error regarding the host of Balssia gasti. Figs 5-12 have not been amended to include non-abbreviated generic names, as this would clutter up the figures needlessly. Although we accept the criticism that a reader not familiar with the generic names would on first glance perhaps have some difficulty, a simple electronic search using the species name in the Suppl File would easily rectify any problems.

I feel that this is a valid comment, as are your responses of Figures 5-12 becoming cluttered with full generic names. Thus, I would like to ask you to state somewhere in the manuscript where these full names can be searched for/found.

Thus, my decision is 'minor revisions' with the emphasis on 'minor'.

---

## Round 0.3 · accepted · Accept

· Academic Editor

Accept

The authors have responded to the last round of minor corrections adequately, and this manuscript is now acceptable to be published.